# Enhancing Emission via Radiative Lifetime Manipulation in Ultrathin InGaN/GaN Quantum Wells: The Effects of Simultaneous Electric and Magnetic Fields, Thickness, and Impurity

**DOI:** 10.3390/nano13212817

**Published:** 2023-10-24

**Authors:** Redouane En-nadir, Mohamed A. Basyooni-M. Kabatas, Mohammed Tihtih, Walid Belaid, Ilyass Ez-zejjari, El Ghmari Majda, Haddou El Ghazi, Ahmed Sali, Izeddine Zorkani

**Affiliations:** 1LPS, Department of Physics, Sidi Mohamed Ben Abdullah University, P.O. Box 1796, Atlas Fez 30000, Morocco; hadghazi@gmail.com (H.E.G.); ahmed.sali@usmba.ac.ma (A.S.); izorkani@hotmail.com (I.Z.); 2Department of Precision and Microsystems Engineering, Delft University of Technology, Mekelweg 2, 2628 CD Delft, The Netherlands; 3Department of Nanotechnology and Advanced Materials, Graduate School of Applied and Natural Science, Selçuk University, Konya 42030, Turkey; 4Institute of Ceramic and Polymer Engineering, University of Miskolc, 3515 Miskolc, Hungary; medtihtih@gmail.com; 5Department of Physics, Faculty of Science, Sélçuk University, Konya 42031, Turkey; walid.belaid@usmba.ac.ma; 6ENSAM, University Hassan-II, Casablanca 20670, Morocco; ezzejjariilyass@gmail.com (I.E.-z.); majda.elghmari-etu@etu.univh2c.ma (E.G.M.)

**Keywords:** quantum wells, radiative lifetime, electromagnetic excitation, impurity, thickness

## Abstract

Ultra-thin quantum wells, with their unique charge confinement effects, are essential in enhancing the electronic and optical properties crucial for optoelectronic device optimization. This study focuses on theoretical investigations into radiative recombination lifetimes in nanostructures, specifically addressing both intra-subband (ISB: e-e) and band-to-band (BTB: e-hh) transitions within InGaN/GaN quantum wells (QWs). Our research unveils that the radiative lifetimes in ISB and BTB transitions are significantly influenced by external excitation, particularly in thin-layered QWs with strong confinement effects. In the case of ISB transitions (e-e), the recombination lifetimes span a range from 0.1 to 4.7 ns, indicating relatively longer durations. On the other hand, BTB transitions (e-hh) exhibit quicker lifetimes, falling within the range of 0.01 to 1 ns, indicating comparatively faster recombination processes. However, it is crucial to note that the thickness of the quantum well layer exerts a substantial influence on the radiative lifetime, whereas the presence of impurities has a comparatively minor impact on these recombination lifetimes. This research advances our understanding of transition lifetimes in quantum well systems, promising enhancements across optoelectronic applications, including laser diodes and advanced technologies in detection, sensing, and telecommunications.

## 1. Introduction

In recent years, semiconductor quantum wells (QWs) have attracted considerable attention from the scientific community due to their potential applications as sources of non-classical light, including the generation of single and entangled photons [1,2]. In particular, quantum wells fabricated from group III-Nitrides (e.g., GaN, InN, InGaN, etc.) have emerged as highly promising candidates for enabling single-photon emission at or near room temperature [3,4]. This is primarily attributed to their significant band offsets, tunable energy band gaps, and exceptional thermodynamic and structural characteristics [5,6,7,8]. With the rapid advancement of growth and fabrication techniques in the realm of semiconductor quantum wells (QWs), it has become feasible to produce high-quality thin quantum wells composed of InGaN/GaN with diverse epitaxial orientations [9,10,11,12]. These factors unequivocally highlight the advantages of employing nitride-based devices as future non-classical light emitters. Nevertheless, in principle, delving into the inherent properties of nitride-based nanostructures allows for an even more extensive exploitation of the potential offered by these systems. For example, in the realm of photonics, InGaN/GaN QWs exhibit remarkable versatility. However, while commercial single-photon detectors typically operate within the blue spectral region [13], InGaN/GaN QWs transcend these boundaries by enhancing various applications, including photodiode detectors and blue laser diodes [14]. Moreover, these QWs find use in devices based on InGaN/GaN multiple QWs, particularly in scintillator and detector applications [15,16]. This adaptability highlights the broad range of photonics applications enabled by InGaN/GaN QWs, making them valuable components for diverse optical technologies.

In the realm of optoelectronics, the radiative recombination lifetime (RRLT) stands as a pivotal parameter, exerting significant influence over key device characteristics such as the open-circuit voltage and light emission rates [17]. The comprehension and fine-tuning of recombination lifetime are fundamental endeavors, critical for enhancing the efficiency and performance of various optoelectronic technologies, spanning from solar cells to light-emitting diodes and photodetectors. The radiative lifetime of electrons and holes has been the subject of thorough investigations, encompassing both theoretical modeling and experimental analysis [18,19,20,21]. In the realm of low-dimensional systems, including quantum wells (QWs) and quantum dots (QDs), the recombination lifetime (RL) of electrons and holes has not received extensive scrutiny and exploration.

Recently, Sun et al. have experimentally measured the recombination lifetime of GaSb/GaAs QD [19]. Li et al. conducted a study on InGaN-based QW exhibiting luminescence in the yellow region of the visible spectrum. They employed both conventional and time-resolved cathodoluminescence techniques. Their findings revealed a consistent increase in the radiative recombination lifetime as the temperature rose, reaching up to 100 K [22]. Aghoutane et al. have recently reported a study investigating the lifetime of exciton carriers in an InAs-based QD with an infinite potential barrier under the influence of varying size, intense laser excitation, and magnetic fields [23]. Im et al. conducted a study on GaInN/GaN quantum well structures grown via LP-MOVPE, employing picosecond time-resolved photoluminescence spectroscopy. Their findings indicate that, at temperatures exceeding approximately 100 K, the decay time exhibits a significant decrease, ultimately reaching approximately 75 ps at room temperature [24]. In a symmetric GaAs coupled quantum well (CQW) heterostructure, Wilkes et al. conducted calculations on spatially indirect exciton states while subjecting them to external electric and magnetic fields [25]. In optical property computations for quantum wells (QWs) and quantum dots (QDs), researchers commonly utilize a fixed radiative recombination lifetime (RRLT), leading to study inaccuracies [26,27,28,29,30,31,32]. Therefore, there is a critical need for a comprehensive investigation of the RRLT under varying conditions, including temperature, pressure, intense laser excitation, and electric/magnetic fields.

To date, there has been limited research on recombination lifetime in the presence of impurities and the combined influence of electric and magnetic fields. Building on our prior investigations into InGaN/GaN heterostructures (QWs) [33,34,35,36], our goal is to explore the dynamic behavior of radiative recombination lifetime under various excitation conditions via theoretical calculations and numerical modeling. Recognizing the limitations of assuming a constant radiative lifetime, our study addresses this knowledge gap via a theoretical and simulation investigation, enhancing the accuracy of optical property analyses in semiconductor materials, particularly in the context of optoelectronic device research.

## 2. Theory and Model

In the context of this study, our primary aim is to undertake a comprehensive examination of the interplay between several crucial factors, including the size, impurity, and influence of electric and magnetic fields, on the radiative lifetime. We consider transitions occurring between electrons occupying the conduction band (CB) and heavy holes situated within the valence band (VB) of a quantum well (QW). To facilitate our investigation, we have designed the quantum well using a non-polar m-plan [$10\bar{1}0$] Wurtzite InGaN/GaN heterostructure [37]. The barrier material (GaN) is characterized by a width denoted as (L). Concurrently, the well region material ($In_{0.1}Ga_{0.9}N$) features a distinct width designated as (l). For visual clarity and reference, a comprehensive schematic representation of the studied system is meticulously illustrated in Figure 1. To investigate the radiative lifetime of the transitions under consideration, we must undertake numerical solutions for the 1D Schrödinger equation, which characterizes the behavior of electrons and holes within the system. This numerical approach is essential due to the inclusion of an impurity term and the consideration of a finite confinement profile, making an analytical solution unfeasible. By solving this equation within the framework of the effective mass theory, we can derive the energy levels and corresponding wave functions for electrons in the CB and holes in the VB. Consequently, we can calculate the transition energy and the dipole matrix element (DME) between electrons and heavy-hole levels.

The comprehensive Hamiltonian, encompassing particle kinetic energy, confinement potential energy, the influence of impurities, and electromagnetic excitation, is expressed as follows:(1)$$H=\frac{1}{2m_{e,hh}^{*}}{\left(P+\frac{e}{c}A\left(r\right)\right)}^{2}+V_{CB,VB}^{e,hh}\left(z,\delta \right)+eFz-\frac{e^{2}}{\varepsilon_{r}^{*}\sqrt{{\left(z-z_{0}\right)}^{2}+y^{2}+x^{2}}}$$
where $m_{e,hh}^{*}\;\mathrm{and}\;\hbar$ are the electron/heavy-hole effective mass within the conduction band and the Planck constant, respectively. $\varepsilon_{r}^{*}$ and e denote the relative dielectric constant of different materials used in this study and the free electron change. $A\left(r\right)$ represents the vector potential of the magnetic field and is given by $A\left(r\right)=\left(\frac{1}{2m^{*}}B\times r\right)$, while $z_{0}$ denotes the impurity location within the structure. To streamline our calculations, we opted for a 1D system, taking into account that in quantum wells, particle confinement occurs solely along the z-axis, which implies y=x=Cte=1.

Incorporating the vector potential term of the magnetic field, the Hamiltonian transforms as follows:(2)$$-\frac{\hbar^{2}}{2m^{*}}\frac{d^{2}\psi_{i}\left(z\right)}{dz^{2}}+\left(\frac{e^{2}\gamma^{2}}{2m^{*}c^{2}}z^{2}+e\xi z+V_{CB,VB}^{e,hh}\left(z,\delta \right)-\frac{e^{2}}{\varepsilon^{*}\sqrt{{\left(z-z_{0}\right)}^{2}+2}}\right)\psi_{i}\left(z\right)=E_{i}\psi_{i}\left(z\right)$$
where γ stands for magnetic field strength, and ξ stands for electric field strength.

To enhance the realism of our study, we consistently employed finite confinement potentials to make this theoretical investigation more representative of real-world conditions. In particular, the confinement potential energy, responsible for confining electrons ($V_{CB}^{e}$) within the CB, and the counterpart for heavy-hole ($V_{VB}^{hh}$) within the VB, are expressed as follows, respectively:(3)$$V_{CB}^{e}\left(z\right)=\left\{\begin{array}{cc} 0, & Well\\ V_{0}^{e}, & Barriers \end{array}\right.$$
(4)$$V_{VB}^{hh}\left(z\right)=\left\{\begin{array}{cc} 0, & Well\\ V_{0}^{hh}, & Barriers \end{array}\right.$$

The energy band gap for an InGaN alloy can be determined via Vegard’s law, which can be expressed as follows:(5)$$E_{g}^{InGaN}\left(\delta \right)=\delta E_{g}^{InN}+\left(1-\delta \right)E_{g}^{GaN}-b\times \left(1-\delta \right)$$

For an indium composition of 10% (δ=0.1) and an experimentally obtained bowing parameter of 2.3 (b = 2.3) [38], the energy band gap of InGaN becomes
(6)$$E_{g}^{InGaN}\left(0.1\right)=0.1E_{g}^{InN}+0.9E_{g}^{GaN}-2.4\times \left(0.9\right)$$
given the energy band gaps of InN and GaN are $E_{g}^{InN}$ (0.69 eV) and $E_{g}^{InN}$ (3.47 eV), respectively [38]. Thus, $E_{g}^{InGaN}\left(0.1\right)=2.976\;\mathrm{eV}$.

$V_{0}^{e}\left(V_{0}^{hh}\right)$ represent the height of the potential barrier within the CB and VB, respectively. $∆E_{g}=E_{g}^{GaN}-E_{g}^{InGaN}$. Thus, $∆E_{g}=0.494\;eV$, $V_{CB}^{e}(V_{0}^{e})=0.358\;eV$, and $V_{VB}^{hh}(V_{0}^{hh})=0.654\;eV$.

The effective masses of the electrons and heavy holes of InN and GaN materials are $m_{e}^{InN}{(m}_{hh}^{InN})\;0.11\;(1.63\;m_{0})$ and $m_{e}^{GaN}{(m}_{hh}^{GaN})\;0.20\;(1.4\;m_{0})$, respectively [39,40,41].
(7)$$m_{e,hh}^{*}\left(\delta \right)=\left\{\begin{array}{cc} m_{InGaN}^{*} & Well\\ m_{GaN}^{*} & Barriers \end{array}\right.$$
(8)$$\varepsilon_{e,hh}^{*}\left(\delta \right)=\left\{\begin{array}{cc} \varepsilon_{InGaN}^{*} & Well\\ \varepsilon_{GaN}^{*} & Barriers\\ \sqrt{\varepsilon_{InGaN}^{*}\times \varepsilon_{GaN}^{*}} & Well/barreir \end{array}\right.$$

To determine the effective masses of electrons (heavy hole) and their corresponding dielectric constants in the ternary alloy (InGaN) representing the confinement region, we employ the same principle—Vegard’s law: $m_{e}^{*}(m_{hh}^{*})$ are ${1.739\;m}_{0}\;(0.174\;m_{0})$. Similarly, the dielectric constant of the InGaN region is $\varepsilon_{InGaN}^{*}(=0.64\;\varepsilon_{0})$, where $m_{0}$ and $\varepsilon_{0}$ are the free electron mass and the dielectric constant of vacuum. The dipole matrix element between electron and heavy-hole states is given as $|M_{if}|=\left\langle \psi_{i}|ez|\psi_{f}\right\rangle$, where $\psi_{i}$ and $\psi_{f}$ represent the initial and final states resulting from the optical transitions (i.e., ISB and/or BTB). Moreover, the transition energy between the allowed states within the system is given as $∆E_{fi}=E_{f}-E_{i}.$ It is important to mention that optical transitions are possible between two different levels only if the selection rule $\left(∆l=\pm 1\right)$ is satisfied.

To determine these primary physical parameters, we have chosen to employ the finite element method (FEM) to solve the Schrödinger equation of the studied system. This numerical technique subdivides complex physical problems into smaller, more manageable elements, employing mathematical principles to approximate solutions within each segment. FEM, widely utilized in engineering and scientific fields, excels in addressing a diverse array of complex challenges spanning engineering, physics, and other domains due to its renowned versatility. Moreover, FEM demonstrates a knack for maintaining both precision and adaptability when modeling irregular geometries and material properties, making it invaluable for solving practical real-world problems [42]. In this study, the presence of a donor impurity within the structure renders the Schrödinger equation inscrutable via conventional analytical means. Consequently, we employ FEM with a one-dimensional mesh (calculation grid) comprising 3N+1 points, where N is set to 50. This approach provides precision for quantum well (QW) systems’ ground and excited states. However, for more complex or higher energy systems, the FEM solution’s precision may decline, requiring a finer mesh and higher-order basis functions. Accuracy depends on factors like problem complexity, numerical parameter choices, and computational resources in contrast to conventional methods like perturbative and variational techniques. Importantly, for the determination of energy levels and their corresponding wave functions, we consider the following boundary conditions [34,39]:(9)$${\left[\underset{n}{\to }.\underset{\nabla }{\to }\left(\frac{\psi }{m_{e,GaN}^{*}}\right)\right]}_{barrier}={\left[\underset{n}{\to }.\underset{\nabla }{\to }\left(\frac{\psi }{m_{e,InGaN}^{*}}\right)\right]}_{well}$$

The system being studied employs a mesh grid consisting of 3N + 1 points. Each layer within the system is discretized with varying step sizes. In particular, the step size for the barriers is labeled as $h_{b}=L/N$, whereas for the regions within the well, it is expressed as $h_{w}=l/N$. Consequently, for k values spanning from 0 to N, the corresponding mesh nodes for a single QW can be determined as follows: the left barrier is located at $z_{j}=k*h_{b}$, the well region is positioned at $z_{j}=L+k*h_{w}$, and the right barrier is situated at $z_{j}=L+l+k*h_{b}$. Utilizing the FEM, we calculate the first and second derivative wave functions [26,43].
(10)$${\left.\frac{\partial^{2}\psi (z)}{\partial z^{2}}\right)}_{z_{k}}=\frac{\psi_{k+1}-2\psi_{k}+\psi_{k-1}}{{\left(z_{k+1}-z_{k}\right)}^{2}}$$
(11)$${\left.\frac{\partial \psi \left(z\right)}{\partial z}\right)}_{z_{k}}=\frac{\psi_{k+1}-\psi_{k}}{z_{k+1}-z_{k}}$$

Suppose that $h_{b}=z_{k+1}-z_{k},$ Equation (4) becomes
(12)$$\left(\frac{-\hbar^{2}}{2m_{e,hh}^{*}}\right)\left[\frac{\psi_{k-1}-2\psi_{k}+\psi_{k+1}}{{\left(h_{b}\right)}^{2}}\right]+V_{0}^{e,hh}\psi_{k}=E\psi_{k}$$

Assuming that $\Omega =\frac{-\hbar^{2}}{2m^{*}{h_{b}}^{2}},$ the same equation above becomes
(13)$$\Omega \left[\psi_{k-1}+\psi_{k+1}+\left(\frac{V_{0}^{e,hh}}{\Omega }-2\right)\psi_{k}\right]=E\psi_{k}$$

The matrix that provides us with the energy levels and corresponding wave functions in this particular region (barrier region) can then be written as follows:(14)$$M_{Barrier}=\left(\begin{array}{cccccc} 0 & 0 & 0 & 0 & 0 & 0\\ \Omega & \left(V_{0}^{e,hh}-2\Omega \right) & \Omega & 0 & 0 & 0\\ 0 & \Omega & \left(V_{0}^{e,hh}-2\Omega \right) & \Omega & 0 & 0\\ 0 & 0 & 0 & 0 & \ddots & \vdots \\ 0 & 0 & 0 & 0 & 0 & \cdots \\ 0 & 0 & 0 & 0 & 0 & 0 \end{array}\right)$$

Remark: similar steps can be followed to obtain the matrix that provides us with the energy levels and corresponding wave functions within the other regions (i.e., well) with the elimination of the potential ($V_{0}=0$), which is zero in none of the barrier regions. The system’s matrix is obtained by summing the three calculated matrices (left barrier, well, right barrier). A code of “Python programming language” has been utilized to perform numerical solutions for these matrices using libraries such as NumPy, SciPy, Math, Matplotlib, and others. Once we have determined the values of $M_{if}$ and $∆E_{fi}$, we can proceed to compute the radiative lifetime between the initial and final states, which arise from optical transitions, such as ISB (inter-sub-band) and/or BTB (band-to-band). Hence, its analytical expression is given as follows [44,45,46]:(15)$$\tau_{fi}=\frac{3c^{3}h^{4}\varepsilon_{0}}{16\pi^{3}∆E_{fi}^{3}n_{r}M_{if}^{2}}$$
where $c,h,\varepsilon_{0},n_{r}$ denote, respectively, the speed of light in vacuum, Planck’s constant, vacuum permittivity, and refractive index relative to the materials used in this study.

## 3. Results and Discussion

In this numerical investigation, we have employed effective units to streamline our calculations. $R_{GaN}^{*e,hh}$ and $a_{GaN}^{*e,hh}$ used in this study are, respectively, the effective Rydberg and Bohr radius at the barrier region (GaN). The effective Rydberg $R_{GaN}^{*e,hh}\left(=\frac{m_{GaN}^{*e,hh}e^{3}}{2(4\pi \varepsilon_{GaN}^{*e,hh}{\hbar )}^{2}}\right)$ is used as unit energy, while the effective Bohr radius $a_{GaN}^{*e,hh}\left(a_{b}^{*}=\frac{4\pi \varepsilon_{GaN}^{*e,hh}\hbar^{2}}{m_{GaN}^{*e,hh}e^{2}}\right)$ is used as the unit of length. The effective electric (F) and magnetic (B) fields are given via the following expressions: For l=4L=4 and δ=10%(0.1), $a_{GaN}^{*e}(a_{GaN}^{*hh})=2.55(3.65)\;nm$ and $R_{GaN}^{*e}(R_{GaN}^{*hh})=0.029(0.203)\;eV$. Therefore, for the electron, $F=\frac{e\xi a_{GaN}^{*e}}{R_{GaN}^{*e}}\;$ and $B=\frac{e^{2}\gamma^{2}a_{GaN}^{*e}}{2m^{*}c^{2}R_{GaN}^{*e}},\;$ while for the heavy hole, $F=\frac{e\xi a_{GaN}^{*hh}}{R_{GaN}^{*hh}}$ and $B=\frac{e^{2}\gamma^{2}a_{GaN}^{*hh}}{2m^{*}c^{2}R_{GaN}^{*hh}}$.

It is important to note that, per Equation (15), the recombination lifetime is directly influenced by changes in both the transition energy and the dipole matrix element. Consequently, any alterations in these physical parameters will certainly impact the behavior of the radiative lifetime. During the discussion of the following results, we will utilize abbreviations for the sake of convenience, particularly when referring to figures and panels. These abbreviations include LSP (left sub-panel), RSP (right sub-panel), and MP (main panel).

### 3.1. Thickness Effect (QW’s Size)

Figure 2 illustrates the variation in transition energy ($∆E_{fi}$), the dipole matrix element ($M_{if\;}^{2}$), and recombination lifetime ($\tau_{fi}^{i};\;i:e\to e,e\to hh$) as a function of well thickness for three different optical transitions in the absence of external excitations, while maintaining a fixed barrier thickness at L = 2. It is obvious that the well’s width has a significant impact on $∆E_{fi}$ (LSP), $M_{if}^{2}$(RSP), and $\tau_{fi}^{i}\;$(MP). Moreover, it is evident that for all transitions, the increase in the well’s width leads to a corresponding increase in $∆E_{fi}$ until it reaches a saturation regime in a weak confinement regime region (l>0.5 nm). This effect is more pronounced for BTB transitions compared to ISB. This increase can be attributed to the reduction in quantum confinement resulting from the enlargement of the well, leading to an enhancement in the excited state energy level ($E_{f}$) relative to the ground state energy level ($E_{i}$). Consequently, $∆E_{fi}$ is expected to increase. Furthermore, it is noted that the transition energy associated with BTB remains consistent within the strong confinement regime (l<0.5 nm) for both considered transitions (i.e, $E_{11}^{e-hh}$ and $E_{21}^{e-hh}$). However, it is evident that $M_{if}^{2}$ demonstrates two different behaviors. It experiences a decrease in the case of thin QW (within the strong confinement regime, where l<0.5 nm) while exhibiting an increase for thick QW (within the weak confinement regime, where l>0.5 nm). This behavior change can be explained by the fact that in the strong confinement regime, there is less overlap between the electron wave functions of the final and initial electron states. In contrast, within the weak confinement regime, the overlap between these wave functions improves. Furthermore, it has been observed that the dipole matrix element related to BTB remains constant within the weak confinement regime (l>0.5 nm) for both of the considered transitions, particularly $M_{11}^{e-hh}$ and $M_{21}^{e-hh}$.

This can be explained by the fact that in this confinement regime, the overlap between the considered electron states ($1s^{e},\;2s^{e},\;1s^{hh}$) changes simultaneously. Additionally, the overlap between ISB transitions is greater than that between BTB transitions, as indicated in the RSP.

As mentioned earlier, it is evident that an alteration in both $∆E_{fi}$ and $M_{if}$ can significantly influence the variations in the recombination lifetime for both ISB and BTB optical transitions. Consequently, it is clear that the radiative lifetime undergoes substantial fluctuations in response to changes in both $∆E_{fi}$ and $M_{if}$ concerning the QW’s thickness. As depicted in the MP, it is readily apparent that the radiative lifetime decreases as the well width increases until it stabilizes in thicker QWs. This reduction occurs more rapidly for ISB transitions compared to BTB transitions. This decline is assigned to the concurrent increase in both $∆E_{fi}$ and $\;M_{if}$ with the increase in the QW’s thickness, which is explained physically by the same reasons discussed earlier in the LSP and RSP. In the realm of quantum physics, we can discern striking disparities in the radiative processes associated with the electron–electron (e−e) transitions under distinct confinement regimes. In particular, when confronted with a strong confinement regime, the radiative timescale $\tau_{fi}^{e-e}$ exhibits a significant variation, manifesting as 4.7 nanoseconds. In stark contrast, the weak confinement regime gives rise to a considerably shorter radiative timescale, with $\tau_{fi}^{e-e}$ diminishing to almost a mere 0.3 nanoseconds. On the other hand, when we turn our attention to transitions involving electrons and heavy holes (e−hh), a divergent behavior is observed. Here, the radiative timescales $\tau_{fi}^{e-hh}$ are substantially faster, characterized by a dynamic range spanning from 0.8 nanoseconds to nearly instantaneous emissions, effectively approaching zero nanoseconds. This marked disparity in radiative behaviors underscores the profound influence of confinement regimes on the radiative properties of quantum systems. These results demonstrate a strong agreement with the existing recent literature on the radiative lifetime of neutral excitons confined in CdSe colloidal nanoplatelets (NPLs) [47], thin GaAs-based quantum box (QB) [48], and GaAs/GaAIAs QW nanostructure at room temperature using the photoluminescence phase shift method [49].

### 3.2. Applied Electric Field Effect (F)

To calculate the exact value of the electric field strength (ξ), we can use the following formula: $F=\frac{e{\xi a}_{GaN}^{e,hh}}{R_{GaN}^{e,hh}}$.

Now, let us, for example, solve ξ for electrons.

Let us suppose that F=1, e (electron charge) $\approx 1.602\times 10^{-19}$ C, and for l=3L=1, $a_{GaN}^{e}\approx 2.56\;nm\;(2.56{\;\times \;10}^{-9}\;m)$ and $R_{GaN}^{e}\approx 29.13\;meV\;(29.13\times 10^{-3}\;eV)$.
$$\xi =\frac{FR_{GaN}^{e}}{ea_{GaN}^{e}}=\frac{1\times 29.13\times 10^{-3}}{1.602\times 10^{-19}\times 2.56\;{\times 10}^{-9}}$$

After calculating this expression, we obtain the real-electric field strength: $\xi =6.13\times 10^{5}\;V/m$.

Figure 3 illustrates the dependence of transition energy, dipole matrix elements, and recombination lifetime on the magnitude of an externally applied electric field, specifically examining various optical transitions within an ultrathin-layered heterostructure composed of InGaN/GaN. The transitions under scrutiny encompass ISB and BTB transitions. Based on the analysis conducted via LSP and RSP investigations, it appears that variations in the magnitude of the applied electric field exhibit distinct behaviors for different optical transitions (ISB and BTB) within the structure, regardless of layer thickness and without applied magnetic field. In particular, for the transitions $E_{11}^{e-hh}$ and $E_{21}^{e-hh}$, there is a clear linear reduction in their energies as the electric field magnitude increases. On the other hand, the transition energy $E_{21}^{e-e}$ exhibits a gradual improvement with increasing electric field strength. Regarding the dipole matrix elements $\left(M_{ij}\right)$, the behavior varies depending on the specific transition being considered. $M_{21}^{e-e}$ and $M_{11}^{e-hh}$ experience a decrease in magnitude with increasing electric field, while $M_{21}^{e-hh}$ exhibits enhancement. This suggests that the degree of overlap between electron and hole wave functions, crucial for radiative processes, can either improve or diminish, contingent on the nature of the transition—whether it involves ISB or BTB transitions. The alteration in transition energies, along with the modifications in electron–hole correlations, certainly exerts a significant influence on the fluctuation of the radiative lifetime associated with the examined transitions; this influence is evident from the MP of Figure 3.

From this panel, it is evident that as the applied electric field strength increases, the radiative lifetime associated with the electron–electron correlation ($\tau_{21}^{e-e}$) initially undergoes a slight increase followed by a gradual decline. In contrast, the radiative lifetime associated with the electron–heavy hole correlation ($\tau_{11}^{e-hh}$) exhibits a smooth increase in response to the electric field magnitude, with a notable acceleration in the improvement of the radiative lifetime pertaining to the transition $\tau_{21}^{e-hh}$. The radiative lifetime ($\tau_{21}^{e-e}$) initially increases until it reaches a maximum value, which occurs at approximately F=0.37, corresponding to a critical electric field magnitude. Subsequently, it gradually decreases. This behavior arises from a competition between two key factors: the variation in the transition energy ($E_{21}^{e-e}$) and the overlap between the electron states ($M_{21}^{e-e}$). As the electric field strength increases, $E_{21}^{e-e}$ experiences an increase, while $M_{21}^{e-e}$ decreases. Consequently, for electric field magnitudes below 0.37, the behavior of the radiative lifetime is primarily governed by the changes in $E_{21}^{e-e}$. Conversely, for electric field values exceeding 0.37, the behavior of the radiative lifetime is predominantly influenced by alterations in $M_{21}^{e-e}$. This critical point at F=0.37 represents a transition regime where the interplay between these factors dictates the behavior of $\tau_{21}^{e-e}$. Moreover, the observed rise in both $\tau_{21}^{e-hh}$ and $\tau_{11}^{e-hh}$ across all electric field values can be attributed to the prevailing influence of the transition energy relative to the overlap between the electron and heavy hole wave functions. This implies that changes in the transition energy ($E_{21}^{e-hh}$ and $E_{11}^{e-hh}$) play a more substantial role in shaping the behavior of these radiative lifetimes in comparison to the overlap effects ($M_{21}^{e-hh}$ and $M_{11}^{e-hh}$). Notably, the pronounced acceleration of $\tau_{21}^{e-hh}$ compared to $\tau_{11}^{e-hh}$ can be attributed to the rapid increase in the dipole matrix element ($M_{21}^{e-hh}$) with higher electric field intensities. This enhancement in $M_{21}^{e-hh}$ as the electric field intensity grows leads to a more significant impact on $\tau_{21}^{e-hh}$ compared to $\tau_{11}^{e-hh}$, contributing to the observed differences in their behavior. Furthermore, it is imperative to emphasize that the radiative lifetime denoted as $\tau_{11}^{e-hh}$ exhibits a notably shorter duration in comparison to the other meticulously scrutinized recombination lifetimes. It becomes evident that in the context of (e−e) transitions, the radiative lifetime $\tau_{21}^{e-e}$ assumes values approximately in the order of 0.25 to 0.20 nanoseconds. In stark contrast, the transitions associated with $\tau_{fi}^{e-hh}$ reveal a distinct behavior, particularly within the domain of weak applied electric fields. Here, we discern that the radiative lifetimes $\tau_{fi}^{e-hh}$ vary within the range of 0.13 nanoseconds to virtually instantaneous emissions, essentially converging toward zero nanoseconds. This conspicuous differentiation in radiative lifetimes underscores the profound influence of applied electric fields on the recombination dynamics within the quantum system under examination. These findings demonstrate substantial alignment with recent studies in the literature that investigate the impact of applied electric fields on recombination lifetimes in nanostructures, particularly a GaAs doped quantum dot (QD) [44].

### 3.3. Applied Magnetic Field Effect (B)

First, let us calculate the value of γ, which represents the real-magnetic field strength in Tesla (T). This can be calculated using the following formula: $B=\frac{e^{2}\gamma^{2}a_{GaN}^{*e,hh}}{2m^{*}c^{2}R_{GaN}^{*e,hh}}$.

Now, let us, for example, solve γ for electrons.

Let us suppose that B=1, e (electron charge) $\approx \;1.602\times 10^{-19}$ C, and for l=3L=1, $a_{GaN}^{e}\approx 2.56\;nm\;(2.56\;{\times 10}^{-9}m)$ and $R_{GaN}^{e}\approx 29.13\;meV\;(29.13\times 10^{-3}eV)$. c (speed of light in vacuum) ≈ 20*m*_0_ (*m*_0_ is the electron mass in a vacuum (=9.10938356 × 10^−31^ kg).
$$\gamma =\sqrt{\frac{B\times 2m^{*}c^{2}R_{GaN}^{*e}}{e^{2}a_{GaN}^{*e}}}$$

Thus, after substituting the numerical values, we obtain the value of $\gamma \approx \sqrt{0.03733}$. So, the value of γ is approximately 0.19 T.

Figure 4 illustrates the intricate interplay between magnetic field strength and various optical properties, including transition energy, dipole matrix elements, and recombination lifetime focusing on ISB and BTB transitions. The figure demonstrates the substantial impact of the applied magnetic field on parameters such as the transition energy difference ($∆E_{fi}$), the dipole matrix elements ($M_{if}$), and the recombination lifetime ($\tau_{fi}$) in relation to the considered optical transitions within the studied system. Eventually, the augmentation of the applied magnetic field produces notable effects on the system’s optical properties. As the magnetic field strength increases, a decrease is observed in the transition energies $E_{21}^{e-hh}$ and $E_{11}^{e-hh}$. This reduction arises from the expanding gap between energy levels $E_{1}^{e},\;E_{2}^{e},$ and $E_{1}^{hh}$, which is a direct consequence of the magnetic field. Additionally, the dipole matrix elements $M_{21}^{e-e}$ and $M_{11}^{e-hh}$ decrease due to the diminishing overlap between the energy levels and their corresponding wave functions under the influence of the magnetic field. Conversely, the magnetic field enhances the transition energy $E_{21}^{e-e}$. These effects collectively highlight the intricate interplay between magnetic field strength and the optical characteristics of the system.

Remarkably, it is worth noting that the dipole matrix element $M_{21}^{e-hh}$ exhibits a bifurcated behavior: it increases for magnetic fields within the lower range (B<0.2) and steadily decreases for an effective magnetic field exceeding 0.2. These factors exhibit sensitivity to alterations in the magnitude of the applied magnetic field, contributing to the observed trends. These trends exert a significant influence on the radiative lifetime within the system, as illustrated in the MP of the same figure. Clearly, as we intensify the magnetic field, we observe a reduction in the recombination lifetime $\tau_{21}^{e-e}$ with a comparatively minor impact on $\tau_{21}^{e-hh}$. However, $\tau_{11}^{e-hh}$ experiences a significant augmentation. Moreover, it is crucial to underscore that $\tau_{11}^{e-hh}$ manifests the shortest duration among the three meticulously investigated lifetimes. We also noted that $\tau_{21}^{e-e}$ experiences variations within the range of 0.05 to 0.24 nanoseconds, reaching its maximum value in the presence of a weak magnetic field. In contrast, $\tau_{fi}^{e-hh}$ exhibits a consistent trend of being approximately zero for all magnitudes of applied magnetic fields. This distinct behavior underscores the profound impact of magnetic field strength on the recombination dynamics within our quantum system. These findings demonstrate substantial alignment with recent studies in the literature that investigate the impact of applied electric fields on recombination lifetimes in nanostructures [50].

Figure 5 illustrates the variation of recombination lifetime associated with ISB and BTB transitions as a function of magnetic field for two distinct electric field values in an InGaN/GaN ultrathin-layered heterostructure, the same investigated system. This figure provides insights into the interplay effect of electric and magnetic fields on relaxation lifetime. However, the introduction and manipulation of an electric field have discernible influences on the lifetimes associated with both ISB and BTB optical transition. In particular, when an electric field (F = 1) is applied, there is a substantial reduction in the lifetime linked to ISB, while the lifetime associated with BTB experiences a modest improvement. It is noteworthy that for the second transition-related lifetime ($\tau_{21}^{e-hh}$), at a critical applied magnetic field value of approximately 0.4 (B ≈ 0.4), the applied electric field ceases to exert any discernible influence on this critical point. This intricate balance and the effects observed in this study underscore the potential for practical real-world applications that can be achieved by mastering the simultaneous manipulation of electric and magnetic fields.

### 3.4. Impurity-Location Effect ($z_{0}$)

To elucidate the influence of impurity localization on the radiative lifetime within the analyzed system, we have compiled the data presented in Table 1. This figure shows the dataset captures changes in impurity positioning within the investigated thin quantum well, devoid of electromagnetic excitation while maintaining a constant nanostructure configuration. The table unmistakably demonstrates that the impact of the impurity presence on $∆E_{fi}$ and $M_{if}$ is negligible, it is approximately on the order of $10^{-4}$. Consequently, it exerts a minimal influence on the modulation of the recombination lifetime $\tau_{fi}$. The impurity was initially located at the left edge of the nanostructure and then shifted to its center; it is denoted as $z_{0}:0\to L+l/2$. It is evident that the energy difference between the electron and heavy hole states has been influenced by the relocation of the impurity within the system. By shifting it from the edge toward the center, $E_{21}^{e-e}$ and $E_{21}^{e-hh}$ experienced a modest reduction, whereas $E_{11}^{e-hh}$ exhibited improvement because of the rapid improvement in $E_{1}^{e}$ and $E_{1}^{hh}$ compared to the energy level $E_{2}^{e}$. Likewise, we observe that shifting the impurity in the same direction has an impact on the overlap between the electron states under consideration. It is noteworthy that both $M_{21}^{e-e}$ and $M_{11}^{e-hh}$ exhibit a marginal decrease initially, followed by an increase as the impurity approaches its central position within the system. This behavior arises from the fact that the electron–impurity correlation is important when the impurity is situated near the interfaces (barrier/well) of the quantum well (QW), which minimizes the overlap between electron–hole wave functions. Consequently, this reduced interaction leads to a decrease in the overlap between electron and heavy hole states in this spatial region.

Nevertheless, it is noteworthy that this influence exerts no discernible impact on $M_{21}^{e-hh}$, as this particular matrix element exhibits continuous enhancements throughout the transitional process. Otherwise, the influence of impurity positioning on the transition energy, dipole matrix elements, and the electron–heavy hole correlation, consequently, affects the radiative lifetimes associated with the ISB and BTB optical transitions examined in this study, as delineated in the third section of Table 1. It is evident from this figure that $\tau_{21}^{e-e}$ and $\tau_{11}^{e-hh}$ initially exhibit a slight increase, followed by a subsequent decline. This behavior can be attributed to the sequential rise and subsequent fall in both the transition energy and the dipole matrix elements associated with these two optical transitions. In contrast, $\tau_{21}^{e-hh}$ displays a marginal increase as the impurity is shifted toward the system’s center. This effect can be attributed to the concurrent increase in the competitive quantities $\Delta E_{21}^{e-hh}$ and $M_{21}^{e-hh}$ that are linked specifically to this transition. To further enrich the scope of this study, our future endeavors are directed toward expanding the investigation of radiative recombination lifetimes. In particular, we plan to explore the simultaneous variation of temperature hydrostatic pressure and intense laser excitation, which will provide a more comprehensive understanding of the underlying mechanisms governing these optical processes.

## 4. Conclusions

In summary, this article is dedicated to a comprehensive theoretical exploration of the electronic and optical characteristics within ultra-thin InGaN/GaN quantum wells. Our primary emphasis is placed on understanding the radiative recombination lifetime related to ISB and BTB optical transitions in the system. This study reveals that radiative lifetimes can be effectively controlled using various strategies, such as size adjustments and electric/magnetic field modifications. Moreover, we observe that radiative lifetimes for ISB transitions are mostly longer than for BTB transitions, and impurity has a minor influence compared to the dominant effects of electromagnetic polarization. These findings hold promise for advancing theoretical calculations related to optical properties in nanostructures, thereby contributing to the advancement of optoelectronics and photonics applications in low-dimensional systems.

## Figures and Tables

**Figure 1 nanomaterials-13-02817-f001:**
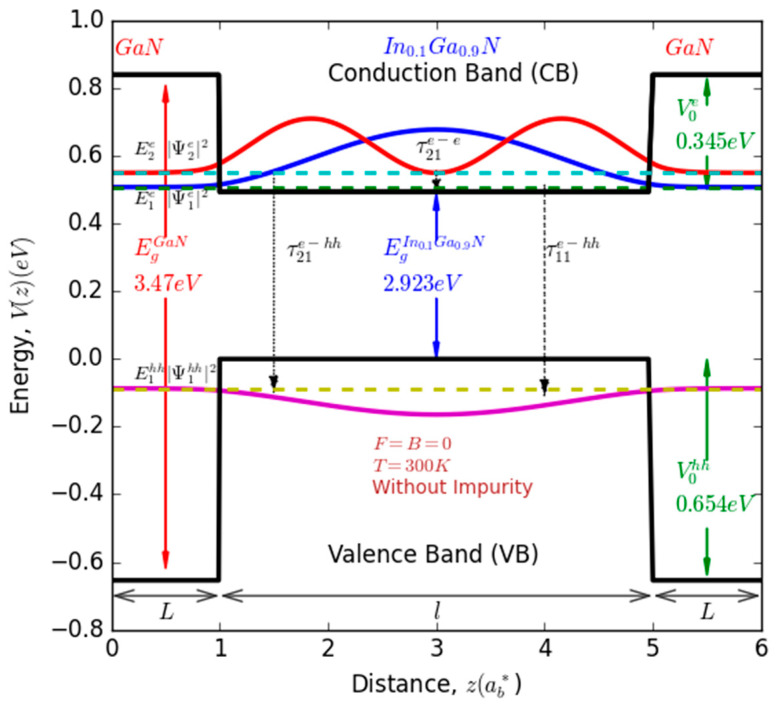
Schematic representation of a single GaN/InGaN/GaN quantum well illustrating various potential optical transitions, their corresponding energy levels, and associated probability density functions.

**Figure 2 nanomaterials-13-02817-f002:**
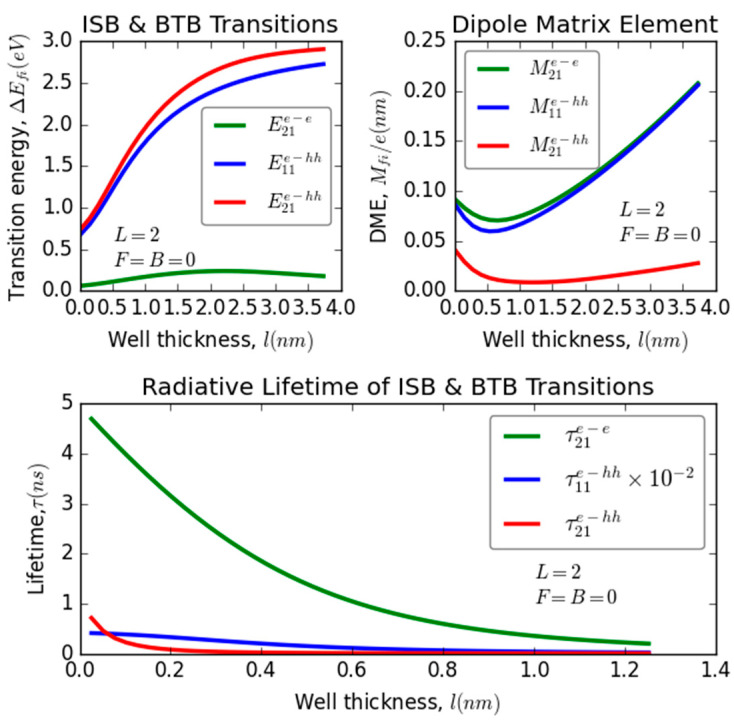
Variation of transition energy, dipole matrix elements, and the recombination lifetime versus the well width for different optical transitions: intra-subband (ISB) and band-to-band (BTB) transitions in InGaN/GaN ultrathin-layered heterostructure (QW).

**Figure 3 nanomaterials-13-02817-f003:**
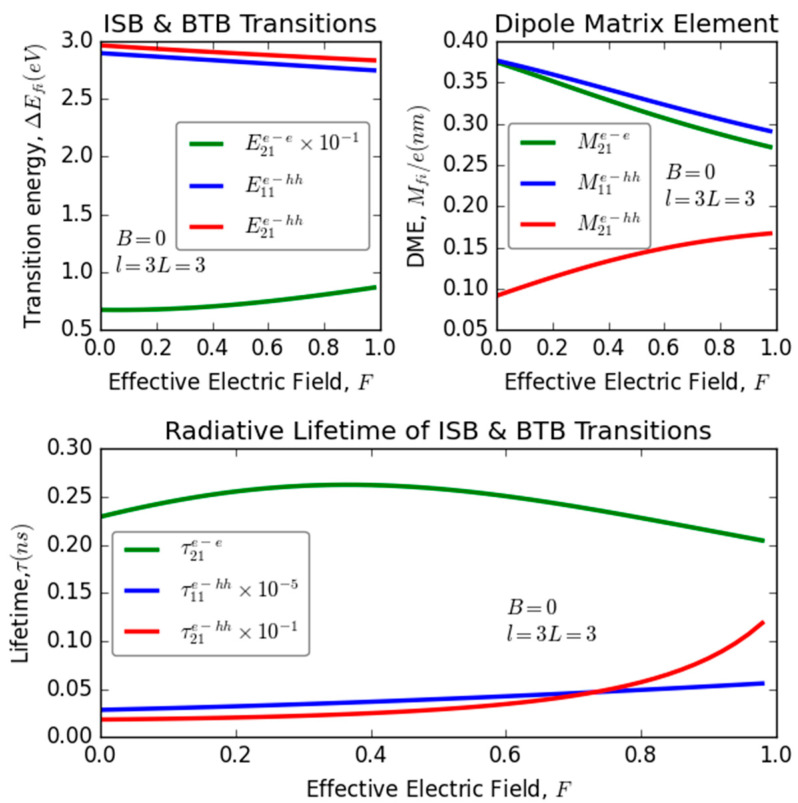
Variation of transition energy, dipole matrix elements, and the recombination lifetime versus the applied electric field for different optical transitions: intra-subband (ISB) and band-to-band (BTB) transitions in InGaN/GaN ultrathin-layered heterostructure (QW).

**Figure 4 nanomaterials-13-02817-f004:**
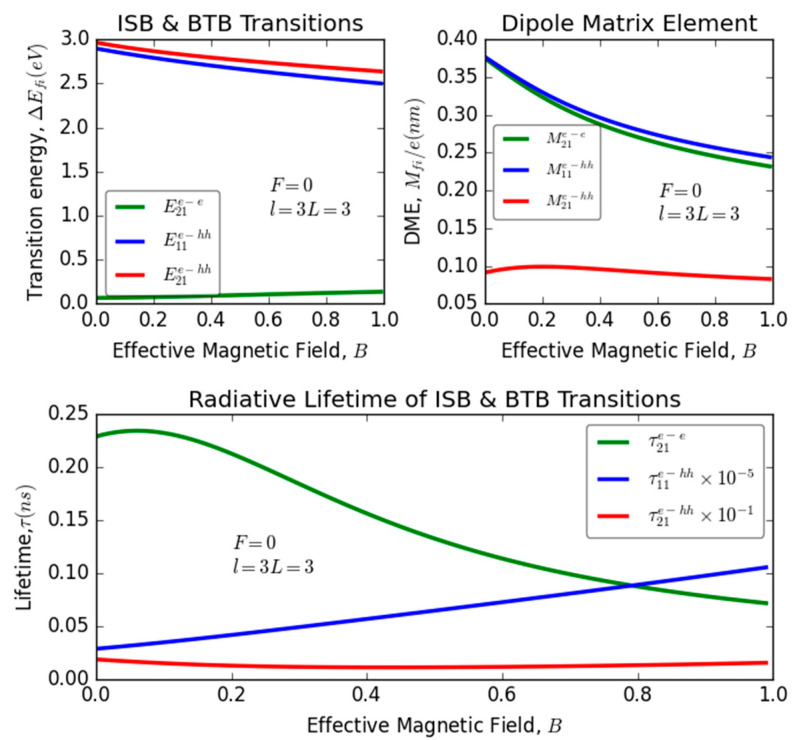
Variation of transition energy, dipole matrix elements, and the recombination lifetime versus the magnetic field for different optical transitions: intra-subband (ISB) and band-to-band (BTB) transitions in InGaN/GaN ultrathin-layered heterostructure (QW).

**Figure 5 nanomaterials-13-02817-f005:**
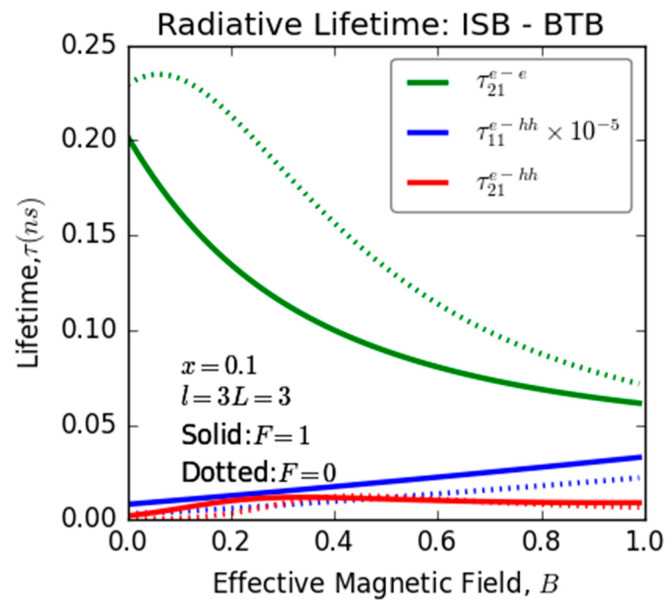
Variation of the recombination lifetime related ISB and BTB versus the magnetic field for two different values of electric field in InGaN/GaN ultrathin-layered heterostructure (QW).

**Table 1 nanomaterials-13-02817-t001:** Show the effect of impurity location on the transition energy, dipole matrix element, and lifetime.

$l=3L=3a_{b}^{*}$ F=B=0	$E_{21}^{e-e}(eV)$	$E_{11}^{e-hh}(eV)$	$E_{21}^{e-hh}(eV)$
$z_{0}$ **(** $a_{b}^{*}$ **)**			
0.0	0.06625	2.89760	2.96386
0.5	0.06610	2.89764	2.96375
1.0	0.06604	2.89767	2.96371
1.5	0.06608	2.89766	2.96374
$z_{0}(a_{b}^{*})$	$M_{21}^{e-e}(\mathrm{nm})$	$M_{11}^{e-hh}(\mathrm{nm})$	$M_{21}^{e-hh}(\mathrm{nm})$
0.0	0.37256	0.37511	0.09271
0.5	0.37224	0.37489	0.09291
1.0	0.37221	0.37486	0.09287
1.5	0.37253	0.37507	0.09254
$z_{0}(a_{b}^{*})$	$\tau_{21}^{e-e}(\mathrm{ns})$	$\tau_{11}^{e-hh}\times 10^{-6}(\mathrm{ns})$	$\tau_{21}^{e-hh}(\mathrm{ns})$
0.0	0.24382	2.8755	0.00170
0.5	0.24591	2.8787	0.00169
1.0	0.24667	2.8791	0.00167
1.5	0.24578	2.8760	0.00164

## Data Availability

Not applicable.

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
