# Peer review of "Enhancing Emission via Radiative Lifetime Manipulation in Ultrathin InGaN/GaN Quantum Wells: The Effects of Simultaneous Electric and Magnetic Fields, Thickness, and Impurity"

_nanomaterials, 2023, doi:10.3390/nano13212817_

Round 1
Reviewer 1 Report
This work presents the theoretical studies of the intraband and band-to-band radiative recombination lifetime in the GaN/InGaN quantum well system. The parameters are set to the well thickness, impurities, and applied electrical field. The results are interesting and conclusive. This manuscript is worthy of publication if the following question is addressed.
1. In the well thickness calculation, the transition energy reduced as the well thickness reduced. In my understanding, the discrete energy levels are increased as the quantum well thickness is reduced due to the strong confinement. Therefore, the BTB transition energy shall rise with the reduction of the quantum well except for the “0” thickness, which is controversial to the results displayed in Figure 2. Can the authors clarify this point?
Author Response
First, we would like to warmly thank you for accepting our invitation to review and for your valuable contribution to improving the quality of our research

Reviewer 2 Report
The manuscript titled "Enhancing Emission through Radiative Lifetime Manipulation in Ultrathin InGaN/GaN Quantum Wells: The Effects of Electric and Magnetic Polarization, Thickness, and Impurity" by Redouane En-nadir et al. is a comprehensive study that effectively examines the impact of various parameters on the emission properties of InGaN/GaN quantum wells. The findings contribute to the existing body of knowledge and have the potential to shape future research in this field. However, a few points could be improved to enhance the clarity and completeness of the presented work.
1. I was unable to ascertain the thickness of the quantum wells used when calculating the influence of electric and magnetic fields on ISB and BTB transitions. This information would be useful to contextualize the discrepancy observed in the Radiative lifetime indicated in Figs 3 and 4, where lifetimes are notably lower at zero electric and magnetic field compared to Fig 2.
2. The manuscript does not explicitly specify the units of the electric and magnetic fields used in the calculations. Providing this information would not only aid my comprehension but also allow for assessing the practical feasibility of these field strengths in a real-world experimental setup.
3. The manuscript lacks discussion on the combined effects of applying electric and magnetic fields on ISB and BTB transitions simultaneously. Incorporating this could offer a more comprehensive understanding of the system under varied conditions and may have practical implications.
————————————
Lastly, setting aside the critique of the manuscript, I'm deeply moved by the authors' sentiments in the acknowledgments. Their aspiration to advance sensing and detection technology for early prediction of natural disasters is genuinely commendable. I extend my best wishes for success in this noble pursuit. My thoughts are with those affected by the tragic earthquake in Morocco.
Author Response
we would like to warmly thank you for accepting our invitation to review and for your valuable contribution to improving the quality of our research

Round 2
Reviewer 1 Report
Thank the authors for the prompt reply to my questions. I am still confused about the same point. So, I would like to spend some time communicating with the authors.
I quote from the authors’s reply, which is “? … calculation of the BTB (Band-to-Band) transition energy is based on the difference between the energy of an electron in the conduction band and that of a hole in the valence band while band gap energy remains constant, ???? = ?? + ?? ?? − ?â„Ž ??. The decrease in the BTB transition energy occurs because the electron’s energy level (?? ??) increases more rapidly than that of the hole (?â„Ž ??), and therefore their difference will increase instead of decreasing. This widening gap between their energy levels is what contributes to the observed reduction in the BTB transition energy as the quantum well thickness decreases.”
I made the points in red. According to your equation, the electron energy level increased more rapidly than the hole. Therefore, the EBTB will be increased instead of decreased. I also think the ?â„Ž ?? rose in another direction in a quantum well, causing the term ?? ?? − ?â„Ž ?? should always be positive.
Author Response
We sincerely appreciate your dedication in reviewing our manuscript.
All authors are diligently addressing your invaluable comments.

Round 3
Reviewer 1 Report
The authors clarify my concerns in detail. I recommend this manuscript for publication.